# Graphene and Reproduction: A Love-Hate Relationship

**DOI:** 10.3390/nano11020547

**Published:** 2021-02-22

**Authors:** Marina Ramal-Sanchez, Antonella Fontana, Luca Valbonetti, Alessandra Ordinelli, Nicola Bernabò, Barbara Barboni

**Affiliations:** 1Faculty of Bioscience and Technology for Food, Agriculture and Environment, University of Teramo, 64100 Teramo, Italy; lvalbonetti@unite.it (L.V.); nbernabo@unite.it (N.B.); bbarboni@unite.it (B.B.); 2Department of Pharmacy, University “G. d’Annunzio” Chieti-Pescara, 66100 Chieti, Italy; fontana@unich.it; 3National Research Council (IBCN), CNR-Campus International Development (EMMA-INFRAFRONTIER-IMPC), Monterotondo Scalo, 00015 Rome, Italy; 4Independent Researcher, Italy; aordinelli@libero.it

**Keywords:** graphene, graphene-derived materials, graphene oxide, reproduction, spermatozoa, in vitro fertilization

## Abstract

Since its discovery, graphene and its multiple derivatives have been extensively used in many fields and with different applications, even in biomedicine. Numerous efforts have been made to elucidate the potential toxicity derived from their use, giving rise to an adequate number of publications with varied results. On this basis, the study of the reproductive function constitutes a good tool to evaluate not only the toxic effects derived from the use of these materials directly on the individual, but also the potential toxicity passed on to the offspring. By providing a detailed scientometric analysis, the present review provides an updated overview gathering all the research studies focused on the use of graphene and graphene-based materials in the reproductive field, highlighting the consequences and effects reported to date from experiments performed in vivo and in vitro and in different animal species (from Archea to mammals). Special attention is given to the oxidized form of graphene, graphene oxide, which has been recently investigated for its ability to increase the in vitro fertilization outcomes. Thus, the potential use of graphene oxide against infertility is hypothesized here, probably by engineering the spermatozoa and thus manipulating them in a safer and more efficient way.

## 1. Introduction

Leafing through the newspapers or magazines, listening to the radio, looking at the television or surfing the Web it is impossible to do not stumble on news regarding the use of graphene in everyday life, from the formulation of new batteries [1] and the construction of windows able to accumulate sunlight for energy purposes [2] to the production of very light tennis rackets [3] and more flexible touchscreens [4].

Graphene is chemically defined as a two-dimensional material or, more impressively, a one-atom thick species formed exclusively of carbon atoms packed in a honeycomb crystal lattice [5,6,7]. Despite it being discovered around the 60–70 s [8], graphene acquired an immense popularity after its first isolation in 2004 [9]. It is characterized by amazing properties, such as strength and elasticity, high electrical and thermal conductivity, mechanical stiffness, transparency, flexibility and high aspect ratio [10,11]. Alongside these own features, the rich family of graphene derivatives constitutes a further source of materials that, depending on their functionalization, acquire complementary properties without necessarily losing those of the graphene progenitor [12]. Among them, graphene oxide (GO) is perhaps the most investigated thanks to its improved solubility features and low-production costs [13]. These special features confer to these carbon-based materials the eligibility to be used for numerous applications in omnifarious fields, from electronic and photonic—such as field emission displays, gas sensors, solar panels or batteries [7,14]—to biomedicine and biotechnology [15,16], with the attainment of rapid DNA sequencing [17], targeted delivery systems [18], tissue regeneration [19] and bioimaging [20].

The spreading of graphene and, overall, its oxidized derivatives into the biomedical research world [21] has led many researchers to contemplate the potential effects derived from their use, with a special interest in the biological interactions between graphene materials and cells, organs or tissues in different animal species [22,23,24,25]. Moreover, the evaluation of the reproductive function is a good tool to elucidate not only the toxicity derived from the use of these materials in the reproductive ability, but also the toxic effects passed on into the next generations. In this way, recently, a few studies have been conducted to describe the effects caused by these materials in germ cells, the spermatozoa and the oocytes, responsible for a successful fertilization and continuation of the species.

The present review is aimed to outline all the research studies focused on the use of graphene and graphene-based materials in the reproductive field, highlighting the consequences and effects reported by the scientific community during the last years. As a first step, a detailed scientometric analysis will be presented, created after analysing the published literature regarding the interaction between graphene and Graphene-Derived Materials (GDMs) and the reproductive function in animal models and humans, with the aim of evidencing the research efforts made on that issue.

Then, particular attention will be directed to the recognition of the various derivatives belonging to the family of graphene-based materials investigated in these researches, with special attention to the oxidized form of graphene, GO. The idea is to follow the classification created by Wick et al. in order to understand structure–activity relationships in the context of health science and safety and to avoid generalizations of the term graphene [26]. In addition, all the toxicological studies published to date regarding the potential effects in the reproductive ability of different animal species either in vivo or in vitro will be exposed, as well as the effects on the progeny. Finally, the relationship between graphene oxide and male germ cells will be extensively described, with special attention to the potential positive effects derived from this interaction, which has been recently described and that could set the basis for the use of graphene oxide in male gametes engineering.

## 2. Scientometric Analysis of Graphene and GDMs and the Reproductive Function

A detailed bibliographic search was carried out to create a data set for the scientometric analysis, made up with the Advanced Search Function of Web of Science Database (WoS) [27].

For the present analysis the topic “graphene” was used in combination with “sperm”, “spermatozoa”, “human spermatozoa”, “oocytes”, “reproduction”, “fertility” and “human fertility”. As a result, 29 documents (26 articles and 3 reviews, no publications emerge before 2012) were found published from 2012 to 2019 (see Table 1). Since data were downloaded until March 31th, 2020, and due to the limitations of the Software in terms of sensibility, two documents were lost and were not included within the analysis: Kong et al., 2019 [28] and Bernabò et al., 2020 [29]. Thus, the total number of documents will reach 31 (28 articles and 3 reviews) even if only those identified by the system were included. Bibliometrix Software (Bibliometrix 3.0 University of Naples Federico II, Naples, Italy) was used to map the analysis [30].

It is evident from the analysis performed that the number of research products is recently increasing (annual growth rate 38.3%), as well as the number of citations (Figure 1), even if, unfortunately, the total number remains very small.

### 2.1. Analysis of the ISI Keywords, Themes and Most Relevant Source

To explore the most relevant issues related to the scientific production in the selected products, the International Scientific Indexing (ISI) keywords were analysed. The words most commonly used inferred from the analysis were: “toxicity” (11), “in-vitro” (8), “nanoparticles” (8), “carbon nanotubes” (7) and “graphene oxide” (6). Moreover, the thematic map showed that “graphene oxide” and “in vitro” are of great importance in the research field studied. Regarding the most relevant source, the analysis showed that the majority of the manuscripts were published in journals included in the Web of Sciences Category (WC-Subject Area) “toxicology”, while only one article was published in a journal within the category “reproduction”.

### 2.2. Scientific Production by Country, Collaborations and Co-Authorship Dynamics

The analysis of the contribution from each country is an interesting tool to evaluate the scientific production. Researchers from China, Italy and Korea represent the most active countries in these topics, representing the 60% of the total of publications. On the other hand, the analysis of the collaborations evidences the absence of strong collaborations among researchers from different countries, with no multiple interactions. The analysis of the network with the dynamics of authors and co-authorships establishes that there are many sub-networks, with all the components characterized by the tendency to form highly clustered structures that do not communicate with each other. Taken together, these data suggest that the scientific community is highly fragmented, highlighting the lack of communication among scientists. Cytoscape Software (3.8.2, Cytoscape consortium) was used to integrate and analyse the networks [31]. Appendix A shows the main topological parameters computed on Co-authors networks with their numeric data. In addition, Appendix A lists all the article used for the scientometric analysis, indicating the authors, journal, year, animal species where the work has been performed (if original research), in vivo or in vitro, the graphene compound used, the doses, if it was a toxicologic study, if it had any detrimental effect and additional comments or information. Highlighted in navy blue the two articles not included in the analysis (probably due to their recent date of publication).

## 3. Graphene family

### 3.1. Graphene Synthesis and Properties

Many methods of preparation have been developed in order to obtain graphene, but they differ in the final size, quality and production costs. In general terms, graphene can be obtained following two different strategies: the top down and the bottom-up. Most mass scale graphene is produced by a top-down approach via graphite exfoliation. This process consists of the peeling off of graphene from graphite, requiring large amounts of solvent and mechanical mixing, sonication or electrochemical treatment [5,32,33,34]. Despite several improvements have been performed in the last years, for example by substituting the organic solvents with surfactants added water [33,35] or supercritical fluids [36], there is still a long way until the obtaining of well exfoliated and defect-free graphene following this technique. Alternatively, the chemical oxidation of graphite to graphene oxide promotes a cheaper exfoliation but it requires harsh oxidizing agents. Additionally, in this case, the subsequent sonication and reduction steps give rise to very defective graphene [33,37]. Several researchers are currently working on that, in order to optimize and improve this process [38].

On the other hand stands the bottom-up synthesis of high quality graphene, exploiting chemical vapor deposition or advanced organic synthetic methods that allows to recover very small amounts of the final material and at very high costs [39,40].

### 3.2. Graphene and Carbon Derivatives

In the present analysis of research studies evidencing the effects of graphene and graphene derivatives on reproductive functions, the biggest problem that came to light, as mentioned above, was the nature and features of the investigated materials. As a matter of fact, the term graphene was often used as a general term to indicate carbon materials spanning from zero-dimensional carbon materials such as carbon black (CB) [41] and nanodiamonds (ND) [42], mono-dimensional carbon materials such as single walled (SWCNTs) and multiwalled carbon nanotubes (MWCNTs) [43]—that can be considered a rolled up form of single or few graphene sheets—and even bidimensional carbon materials such as graphene itself.

In order to shed light and clarify the different materials investigated and generically referred to as graphene or carbon materials, the following is a list of the most representative materials that have been encountered during the present bibliographic analysis.

#### 3.2.1. Carbon Nanotubes

Pristine and acid purified SWCNT, differing for their metal (iron) content [44] and oxidized SWCNTs, i.e., SWNTs functionalized with carboxylic moieties [45] have been considered. Nevertheless, the comparison between these studies is not easy because these carbon nanotubes have been characterized only in terms of length, i.e., 100–1000 nm and 1–5 µm (depending on the concentration), respectively. Similarly, when MWCNTs in their pristine forms [46] or oxidized version [47] have been investigated, the Authors characterized the nanotubes according to different features, i.e., the outer diameter of 10–20 nm for MWCNTs and 10–50 nm for oxMWCNTs, length of 10–30 μm for MWCNTs and <5 μm for oxMWCNTs and ζ-potential of −24.7 ± 0.8 mV for MWCNTs and −34 ± 1.6 mV for oxMWCNTs. It is also known that the intensity ratio of D and G bands in Raman spectroscopy, I_D_/I_G_, index of occurred oxidation, is 1.55.

#### 3.2.2. Graphene Nanoplatelets or Graphene Nanoparticles

Graphene nanoplatelets or graphene nanoparticles (GNP) in their pristine version [48,49] or functionalized analogues [48] have been scarcely characterized in the above mentioned papers reporting only their dimensional features and thus hindering the comparison of the evidenced results with new data.

#### 3.2.3. Graphene Quantum Dots (GQDs)

Despite the reduced molecular weight of such material, only a few studies reported detailed properties and dimensions of GQDs, although bottom up and top-down approaches of synthesis could entail completely different products. For example, whereas a commercial sample was described only in terms of round shape of 40–45 nm size and X-ray diffraction pattern [50], other studies reported detailed information regarding dimensions, ζ-potential, XPS with elemental and functional content, IR and Raman analyses [51]. Only such a detailed characterization allows the comparison of new data with published results.

#### 3.2.4. Nitrogen-Doped Graphene Quantum Dots (N-GQDs)

These particles are a variant of pure carbon quantum dots that contain also nitrogen and oxygen. They are prepared by different methods: solvothermal approach in DMF using GO as a precursor [52] or pyrolysis of L-glutamic acid powder at 250 °C (bottom-up method) [51] or cutting down GO (commercial, top-down method), to name a few examples. Characteristically, they emit fluorescence exhibiting an emission peak in the UV-vis region.

#### 3.2.5. Graphene Oxide (GO)

Graphene Oxide (GO) is typically produced, as reported above, by a Hummer’s method [53] or slight modifications of it, through the oxidative exfoliation of graphite using KMnO_4_/NaNO_3_ and H_2_SO_4_ [54]. The product, graphite oxide, is sonicated in order to obtain well exfoliated graphene oxide. Therefore, graphene oxide is a single atom thick molecule containing graphene like areas formed by *sp^2^* carbon atoms and oxygen-functionalized regions containing moieties such as hydroxyl (–OH) and epoxide (–O–) functional groups on the two accessible sides, and carboxylic acid (–COOH) or carbonylic (C=O) groups at the edges. It is a hydrophilic derivative of graphene and possesses different degrees of oxidation depending on the precise protocol of preparation [55]. The graphene-like areas of GO containing free π electrons are hydrophobic and capable of drug loading and non-covalent surface modifications by π–π stacking and van der Waals interactions. Interestingly, the epoxide, hydroxyl, carbonyl and carboxylic acid groups confer to GO a polar character and, although not necessarily charged, allow electrostatic interactions, hydrogen bonding and relevant reactivity [20]. Research studies reporting GO as the investigated carbon nanomaterial are numerous and therefore its features are indispensable to properly evaluate and compare its effects and results. The majority of GO is synthesized by using the Hummer’s method [46,47,56,57,58] even though sometimes this information is omitted (overall in the case of commercial samples). However, the use of this synthetic methodology does not ensure a precise characterization of the final product that may differ in terms of exfoliation degree (single, SLGO, or few layered sheets, FLGO) [48], oxygen content (either in terms of C/O *w*/*w*, that may vary from 0.25 [59] to 0.39 [48] or as final oxygen weight with respect to the total weight, i.e., 20% [41] or >36% [60]), dimensions of the sheets (i.e., from 297 nm mean size [61] to 3–5 µm [41,47]), surface charge (i.e., from ζ -potential −14.13 ± 11.1 mV [60] to −56.7 ± 1.5 mV [46]). In a few studies GO sheets are named nano-GO (nanoGO or NGO) [62,63] or are distinguished in small GO (S-GO) and large GO (L-GO) [64] sheets when the protocol of preparation involves a further step of cutting into small pieces by sonication [64] or harsh oxidation conditions.

#### 3.2.6. Functionalized GO Materials

All materials obtained by reaction of GO with different agents can be inserted in this class. For instance, carboxylic GO (GO-COOH) can be obtained through oxidization on GO surface of epoxy and hydroxyl groups to carboxyl groups by performing chemical modification with sodium chloroacetate [56]. Within this group stand up polyethylenimine functionalized GO (GO-PEI) and chitosan-functionalized GO (GO-CS) as well, both synthesized using carboxyl activating reagent (EDC) to activate the formation of an amide linkage between GO and PEI or CS, respectively [65].

#### 3.2.7. Reduced GO (rGO)

This compound is a reduced version of graphene oxide, often named as pristine graphene due to the fact that, theoretically and as reported above, the reduction in oxygenated moieties of GO should bring graphene oxygen to its original progenitor, graphene. Obviously, this is not perfectly true and rGO keeps numerous defects and *sp^3^* carbon atoms, missing in pure graphene. rGO can be obtained following different approaches. The widely used reductive agent is hydrazine [45,57,66] but hydrothermal treatment, polyphenols or citric acid have been as well investigated [57]. Similar to GO, and depending on the precise protocol of preparation, rGO can have different features in terms of dimensions, oxygen content and exfoliation degree.

## 4. Graphene and Reproduction: Toxicological Studies

In recent years, the spread of graphene-derived materials has caused many researchers concern regarding the environmental and health impacts derived from their use, giving rise to different manuscript publications.

As has been inferred from the scientometric analysis, the majority of the studies conducted to date regarding graphene and GDMs and reproduction have aimed to analyse the toxicological events involving the reproductive function, a powerful approach that allows examining not only the detrimental effects caused to the single individual but also the effects left in bequest to the offspring. In that way and from 2014 (see Figure 1), some researchers have focused their works on elucidating the effects induced directly to the developing embryos in different species [67,68,69,70]. Zebrafish, an aquatic vertebrate useful to assess toxicity in aquatic environments, is one of the species most frequently used to this aim. For instance, Liu and co-workers incubated zebrafish embryos in their first 96 h with different carbon-based compounds (MWCNTs, GO and rGO) at concentrations of 1, 5, 10, 50 and 100 µg/mL, showing low toxicity with some sublethal effects based on the heart and hatching rates and the length of the larvae, more evident in the rGO-exposed embryos [46].

Other researchers, by contrast, have carried out their experiments in different animal species and either in vivo or in vitro by analysing the health of the offspring after exposing the forefathers with different graphene derivatives and concentrations, as it will be further reviewed (summarized in Figure 2).

### 4.1. In Vivo Studies

Different animal species have been used to evaluate the in vivo toxic effects derived from the use of GDMs, from non-mammals (nematodes, insects, fishes) to mammals (mice, rats).

#### 4.1.1. In Vivo Studies Using Non-Mammal Species


*Caenorhabditis elegans and paracentrotus lividus*


The soil nematode hermaphrodite *Caenorhabditis elegans (C. elegans)* is one of the most common species used to study the toxicological phenomena derived from the use of non-physiological compounds, thanks to its relatively short life, low cost, well-known genetic background and easy handling. In this regard, various graphene-derived compounds at specific concentrations have been added by some research groups to the soil living-medium, showing low or no toxicity [28,49,52] or dose-dependent negative effects for the reproductive ability and the offspring [48,71]. For instance, Zanni et al. incubated the worms with large graphene nanoplatelets, GNPs (average size 9 µm and 9 nm thick), for 3 h at doses of 50, 100 and 250 µg/mL to illustrate that their reproductive capability was unaffected by the use of this graphene material, measuring the viability and the brood size [49]. Similar results were obtained by Kong and collaborators using graphene and a mixture of graphene and poly-lactic acid (PLA), a biodegradable thermoplastic polymer easy to combine with graphene. Graphene is widely used in order to increase the mechanical properties of the polymer. In this case, PLA was used here to favor graphene biocompatibility (PLA-graphene). To evaluate its toxicity in vivo, PLA-graphene was incubated at different doses (50, 200, 500, 1000 µg/mL) with the nematodes up to 48 h, revealing the absence of significant changes in the number of the offspring produced by the graphene-treated worms [28]. It is worth noting that no features of graphene or PLA-graphene were reported in the paper.

Zhao et al. investigated a completely different material, nitrogen-doped GQDs characterized by the presence of both nitrogen and oxygen moieties and by a really small size (i.e., average diameter of 3 nm and thickness of 0.5–1 nm). They were used to evaluate the potential toxic effects at different concentrations (0.1, 1, 10 and 100 mg/L). Authors demonstrated the absence of toxicity for the nematodes, targeted organs and embryos [52].

On the other hand, Chatterjee and co-workers have found some negative effects in the reproductive potential of the worms influenced by the surface functionalization and layer number of the different graphene materials utilized [48]. Concretely, they used single-layer graphene oxide (SLGO), few layers graphene oxide (FLGO) and three functionalized types of GNPs: pristine, CONH_2_ (amide) and COOH (carboxylate) at various concentrations (5, 10, 20, 50 µg/mL) for 72 h. In this case, GO and GNP were precisely characterized, being essentially micrometric flakes, not necessarily single layer. Their findings, based on the number of offspring at all stages beyond the egg, showed some toxic effects varying with the graphene type, surface functionalization, number of layers, dose and time exposure, with no toxicity for the lowest concentrations (5 and 10 µg/mL). In a later work, they further investigated and compared the effects of few-layered nano-sized GO and rGO at doses of 20, 50 and 100 µg/mL for 72 h, demonstrating a higher reproductive toxicity of GO than rGO, likely due to the activation of a non-canonical Wnt-MAPK signaling cascade as the possible underlying mechanism causing the GO reduced reproductive capability [71].

The toxicity caused directly to the reproductive organs (concretely the gonads) has been also evaluated. Zhao and collaborators added GO at different doses (1, 10, 100, 1000 µg/mL) to the worms’ living medium until their young adults stage. In this case, GO was synthesized from graphite and had a size of 40–50 nm, prevailingly monolayered. They found that concentrations of GO 10–100 µg/mL induced the production of more germ cell corpses (thus inducing germ line apoptosis) as well as a slightly decreased brood size and number of oocytes, an inhibition of the egg injection rate and an increased embryo mortality. Moreover, they suggested an epigenetic protection mechanism probably activated by GO, hypothesizing a novel self-protection mechanism against toxicity [58].

After long-term GO exposure (72 h, at a concentration of 10 µg/mL) of young adult nematode, GO-characteristic Raman spectral bands measured throughout the bodies of C. elegans revealed the accumulation of GO in the reproductive organs, a reduced sperm number and a damage of the spermatogenesis [61]. Additionally, in this case, commercial GO samples were few-layered, nano-sized flakes with a hydrodynamic radius of 297 nm.

The Mediterranean purple sea urchin (*Paracentrotus lividus*) has been also used by researchers to study the sperm toxicity after the exposure to commercial graphene materials, such as carbon black (CB) and GO. As far as GO is concerned, it was prevailingly mono-layered with a mean size ranging between 0.5–5.0 µm and containing 20% by weight of oxygen. Their findings showed that, after 1 h of sperm exposure to different concentrations (0.0001, 0.001, 0.01, 0.1 and 1.0 mg/L), egg fertilization was significantly affected with a reduction of around 50% in the case of CB or had almost no effect in the case of GO [41].


*Acheta domesticus, enchytraeus crypticus and spodoptera frugiperda*


Another way to study the toxic effects produced by the graphene-derived compounds is administering them with the food and/or water, as has been done by Dziewiecka et al. to evaluate GO toxicity in the *acheta domesticus* species. In this way, the Authors tried to decipher the long-term toxicity in house crickets by analyzing their reproductive capability after nourishing the insects with single-layered micrometric GO at concentrations of 20 and 200 µg/g of food over two generations. Their results showed a reduced fertility, based on the number of laid eggs per female and larvae hatching rates [59]. The same research group reported the effects derived from the use of much smaller nanodiamonds, similarly added to the animal’s diet, finding similar effects but less severe and limited with respect to the GO of the exposed generation [42].

Similar experiments were performed with the soil invertebrate *Enchytraeus crypticus (E. crypticus)* to evaluate the effects of commercial GO and rGO at concentrations of 250 and 1000 mg/Kg of dry soil. By evaluating the life cycle of the soil, the Authors confirmed reduced hatching, survival and reproduction rates after the treatment with the highest concentrations and in a dose-dependent way [60] for the sole GO samples and correlated this effect to the oxidation degree of graphene. GO and oxMWCNT were also supplied with the diet of the generalist insect *Spodoptera frugiperda (S. frugiperda)* by Martins et al. at different doses (10, 100, 1000 µg/g of food). Both monodimensional and bidimensional carbon nanomaterials affected the reproductive performance when they were used at the highest concentrations, which could entail ecologic and economic consequences for the agriculture sciences [47].

#### 4.1.2. In Vivo Studies in Mammals


*Mouse*


Mouse species is commonly used for the study of reproduction, since mice are cheap, easy to maintain and straightforward to breed in captivity [72]. Additionally, the use of mice allows evaluating not only the in vitro fertilization (IVF) rates and first phases of the embryo development, but also the health of the resultant offspring and their potential epigenetic modifications. It is important to consider that the mice model is currently used for the study of the majority of the human diseases, due to the scant complexity to genetically manipulate them and the great similarities with humans, sharing virtually the same set of genes. For these reasons, many Authors have used this model to analyse the detrimental systemic effects caused by graphene materials, mostly after intravenous (IV) injection to both females and males and with special attention to the potential disruption of the reproductive capability. Xu and co-workers examined two sizes of rGO (small, 20–150 nm and large, 200–1500 nm) at different doses (6.25, 12.5, 25 mg/kg mouse) by IV injection into female mice. The experiment was performed at Day 1 or Day 30 prior to the mating or during the pregnancy (early and late). Their results showed that the oestrogen levels, the mating behaviour and the pups (until the third litter of offspring) were unaffected. However, some adverse consequences were found when the treatment was done during the pregnancy, based on the number of abortions or deformed foetuses, with no secondary effects to the later generations [73]. No difference between small and large flakes were evidenced. Similar studies were conducted by Zhang et al. but using a different graphene compound, GQDs labelled with Cu^2+^, to analyse GO biodistribution. Male mice were exposed to this compound by oral gavage (doses of 60, 100 and 300 mg/kg mouse) or IV injection (doses of 25, 75 and 150 mg/kg mouse) to study the short and long-term reproductive ability. Their results illustrated that the exposed mouse possessed normal sexual behaviours with no affectation of the testes and epididymis, normal pregnancy rates and litters, and a normal development and growth of the offspring. GQD-exposed mouse sperm retained high quality, suggesting that GODs avoided to penetrate the blood-brain and blood-testis barriers [51]. On the contrary, using raw (R-SWCNT) and acid purified (P-SWCNT) single walled carbon nanotubes (50 and 100 µg/kg mouse, instilled intratracheally) Park et al. illustrated an immunological response versus Th-1 (inflammatory processes) and a decrease in the pregnancy rates, overall with the highest concentrations [44]. Nevertheless, they did not evidence any dependence of the monitored effects on Fe content, whereas metallic impurities are widely considered the main factor determining toxicity after the exposure to SWCNTs.

Akhavan et al. performed a work in which they confirmed the uptake of NGO sheets by the testis of male mice after IV injecting the compound to both female and male mice in vivo every week for 8 weeks at concentrations of 2, 20, 200 and 2000 µg/mL (last one corresponds to 4 mg/kg mouse). They used GO nanosheets (lateral dimensions < 100 nm) obtained by oxidation and sonication from GO. They analysed the spermatozoa, finding a reduced viability, motility and progressive motility and a decrease in some kinetic parameters, as well as a higher reactive oxygen species (ROS) production and DNA fragmentation when exposed to the highest NGO concentrations. In addition, they reported for the female mice a decreased fertility, gestation rates and hormone concentrations (follicle stimulating hormone, luteinizing hormone, progesterone, prolactin) when using the highest concentrations of this graphene material [62].

Similar nanoGO samples were used by Liang and co-workers to perform their experiments, but results were completely different from the previous study. Concretely, they used two sizes of NGO previously used for cancer phototherapy and in vivo tumour targeting [74,75] (54.9 ± 23.1 for small-sized GO, S-GO and 237.9 ± 79.3 nm for large-sized GO, L-GO, thickness < 4 nm) injected at different concentrations (120 and 300 mg/kg for S-GO and 120 mg/kg for L-GO, administered every 24 h for 5 days) into the tail veins or abdominal cavities of male mice. After 30 days, they evaluated the health of the spermatozoa, the activities of some epididymal enzymes and the sperm function by analysing the litter (pups number, sex ratio, weights, pup survival rates or pup growth over time). Their findings reveal an absence of toxicity even when using the highest concentrations (up to 300 mg/kg), which the Authors explained with the presence of blood-testis and epidydimal barriers in the male mice that will prevent the entry of big-size molecules into the testis [64]. In particular, the high concentration of GO used in the latter study could favour the formation of protein-GO aggregates that reduced further their tissue accumulation, favouring GO clearance [76]. Moreover, as far as the biodistribution analyses are concerned, while in the former study of Akhavan et al. the biodistribution was monitored by functionalizing the NGO with PEG and thus favoring a higher persistence of nanomaterial in the blood circulation, in the latter study of Liang et al. they chose to label NGO with ^125^I, whose stability is known to be not very good for long-term tracking [76].


*Rat*


Nirmal and collaborators used Wistar rats to evaluate the toxicity of nanoscale GO (NGO, 5–10 µm sized, few layered sample) intraperitoneally injected at doses of 0.4, 2 and 10 mg/kg for 7, 15 or 30 days (injected on alternate days). Their findings demonstrated a dose-dependent reduction in the number of spermatozoa, spermatogonia and spermatids, a decreased sperm motility, and some morphological abnormalities in the groups that were treated with the highest NGO concentrations. Moreover, some oxidative stress was confirmed with the increased activity of antioxidant enzymes, without causing damage to the testis or reducing the fertility potential, data confirmed with the healthy offspring derived from the matching with female rats [63].

### 4.2. In Vitro Studies

The study of the toxicological effects in vivo provides an interesting amount of information about the general and systemic effects, as well as the evaluation of many vital functions, organs and tissues as a whole. However, this model poses an important problem, since it acts as a black box: it is possible to know the details about the input (the administration of graphene materials) but not the output (all the single events that lead to the appearing of infertility, for instance). Moreover, it is of great importance to consider the neurohormonal control of reproduction, involving a network of central and peripheral signals in the hypothalamic-pituitary-gonadal axis [77]. Due to the enormous complexity of the system and the wide variety of interactions and effects to study, in vitro studies are necessary to complement the information obtained from the in vivo study and isolate the events for a clearer understanding. Following this approach, it is possible to exert a higher control and analyse the effects from a physiological, histological and molecular point of view. Specifically, the evaluation of the germ cells, the spermatozoa and the oocytes, emerge as a great strategy to analyse the biological, physical and chemical interactions between the reproductive cells and the graphene materials, allowing us to decipher all the consequences of this interaction in a more direct and less intricate way.

To our knowledge, only one research work has been published to date regarding the toxicity of graphene on the female germ cell, the oocyte. Lin et al. recently incubated in vitro GDQs with mouse oocytes at doses of 0.5, 1 and 1.5 µg/mL. By evaluating the oocyte maturation at different times, they found an interference in the maturation process, with no appreciable toxicity regarding the development of the offspring in vivo [50]. Further experiments should be done to study the interaction between the oocyte and graphene materials and its potential effects and consequences.

Regarding the toxicity of the graphene materials in vitro caused to the male germ cells, a modest number of articles have been published to date. Asghar and co-workers functionalized carbon nanotubes (SWCNT-COOH) and rGO to evaluate their influence on human spermatozoa when incubated with different concentrations (1, 5 and 25 µg/mL) over 30 min and 3 h. The Authors found an absence of toxicity on sperm viability and kinetic parameters, but an increase in ROS production when using SWCNTs-COOH, supporting the possibility to develop microfluidic sorting systems to select the sperm subpopulations [45].

Hashemi and co-workers found some cyto- and genotoxic effects in mice spermatozoa when exposed to GO and various rGO: GO reduced with hydrazine (N_2_H_4_-GO), reduced by hydrothermal reaction (HT-GO), and with green tea polyphenols (GTPs-GO). After incubating for 2 h with different doses (0.1, 1, 10, 100 and 400 µg/mL), they found a reduced viability, motility and progressive motility whereas some kinetic parameters, for concentrations >1 µg/mL, were influenced in a dose-dependent manner. Moreover, all the compounds but the GTPs-GO increase the ROS and nitric oxide (NO) production, while decreasing the ATP and NAD+/NADH [62]. However, and as it will be further discussed below, the reduction in the sperm motility and kinetic parameters could be explained by the attachment of the spermatozoa to the GO and rGO sheets, intercepting the sperm free movement but without interfering with the sperm fertilizing ability, as our group could state from the experiments performed [56].

By using GO obtained by a modified Hummer’s method of oxidation, our group was able to have plenty of a cheap, well characterized, clean of metal impurities, easy to be reproducibly obtained with the elected dimensions and oxygen content.

Our main motivation for evaluating the GO effects on spermatozoa was, despite the above few mentioned papers referring to sperm cell toxicity, the lack of works evaluating the potential toxic effects derived from the use of graphene compounds on spermatozoa during the process of capacitation. Capacitation is defined as a series of events that spermatozoa undergo to acquire their fertilizing competence, thus allowing them to fertilize the female egg. This process is absolutely mandatory and can be achieved both in vivo within the female tract [78] or in vitro, by adding some specific components (BSA, bicarbonate) to the capacitating culture medium [79]. On this basis, our group focused its first analysis on the evaluation of GO toxicity on male germ cells in a swine in vitro model and under capacitating conditions. This model was selected due to the possibility to use a protein-free system, an important question since it has been reported [65,80] that GO is able to interact with albumin and other proteins contained in the culture medium, interfering with the bioavailability of the GO sheets. Moreover, the use of this animal model allows the translation of the results into human and study some pathologies related to human infertility, due to the proximity of both species. In that way, spermatozoa were cultured in a capacitating in vitro medium with different GO concentrations (0.5, 1, 5, 10 and 50 µg/mL) and evaluated at different times (30 min, 1 h, 2 h, 3 h, 4 h) [56].

Thanks to this first work it was possible to establish the concentrations of 0.5 and 1 µg/mL as biologically relevant without causing any negative effect, while concentrations ranging from 5 to 50 µg/mL significantly interfered with the acrosome integrity, causing the loss of the acrosome membrane and thus damaging the sperm function. Indeed, by performing an IVF assay with in vitro matured oocytes and spermatozoa treated with GO, an increased number of fertilized oocytes was evidenced when the incubation was performed with GO 0.5 and 1 µg/mL, improving in this way the efficiency of the system. Interestingly, the sperm membrane was analysed by atomic force microscopy, standing out this site as the site of the interaction between GO and the male gametes.

Among the events enclosed within the process of capacitation it is worth mentioning the increase in the membrane fluidity due to an extraction of the cholesterol present in the membrane. Since it is well accepted that spermatozoa are transcriptionally silent (although with controversy [81]), most of the biochemical events take place or initiate within the sperm membrane. Our group was able to confirm, in a later work [82], the cholesterol extraction from the sperm membrane by the action of GO, thus increasing the sperm membrane fluidity and contributing to the acquisition of the fertilizing ability.

In order to confirm the positive results obtained in the IVF assays in a different animal model and in more physiological conditions, further experiments were carried out using a bovine in vitro model in which a somatic component, the oviductal epithelial cells, was included. This innovative model allowed us to verify the absence of toxicity when spermatozoa were capacitated in the presence of GO 1 µg/mL, to confirm the positive effects in terms of IVF outcomes, and even more important to evidence the absence of toxicity during the early embryo development (until Day 8 post-fertilization). Furthermore, the increase in the bull sperm fluidity was confirmed, giving rise to an interesting number of proteins and lipids modified after the treatment with the GO solution [83].

Very recently, our group moved forward towards the use of the mouse model, currently used for the study of most of human diseases since mice and humans share virtually the same set of genes (almost 4000, from the genes studied to date) [84]. The results showed that the addition of 0.5 μg/mL GO to a sperm suspension before performing IVF was able to increase both the number of fertilized oocytes and, surprisingly, the birth rate after embryo transplantation in foster mothers, in a more effective way than the gold standard in promoting in vitro fertility of mice spermatozoa (methyl-β-cyclodextrin). Furthermore, it was demonstrated that GO exerts its positive effect by extracting cholesterol from the membranes, without affecting the integrity of the membrane microdomains and thus preserving the sperm function [29].

## 5. Graphene Oxide: Towards Male Gametes Engineering?

As has been exposed, the vast majority of the works published to date related to graphene and reproduction are focused on the study of the toxicological events that this fusion entails. However, the first approach of what might be the first steps into the bioengineering of the male gametes to minimize the problem of infertility has also been presented, a worldwide issue that affects around the 20% of couples in the world (48.5 million) and entails psychological, social, demographic and financial problems for the society. It is known that the male factor is responsible for around the 50% of the cases, and among them, the 30% are due to unknown causes [85,86,87]. This is the reason that has encouraged many infertile couples to turn to the assisted reproduction technology (ART), especially IVF and the most invasive intracytoplasmic sperm injection (ICSI) techniques. Since the success of this techniques lies at around 26–29%, there is an urgent need to improve their outcomes, mostly for IVF.

It is important to mind that, as reported by the Grand View Research Incorporation (https://www.grandviewresearch.com/press-release/global-assisted-reproductive-technology-market) “*the global assisted reproductive technology (ART) market is expected to reach USD 45.4 billion by 2025, according to a new report by Grand View Research, Inc. Childbearing postponement is one of the high impact rendering drivers of the infertility treatment market. The reproductive behavior and pattern of some of the nations such as few European nations have shifted from early to late childbearing pattern. Increasing marital age, rising tobacco and alcohol consumption, increasing obesity rate are some of the other factors contributing to the market growth. Furthermore, increasing incidence rate of conditions such as poly-cystic ovarian syndrome (PCOS), tubal factors and endometriosis are other drivers of the market”.*

## 6. Conclusions

In summary, an updated overview of the scientific literature concerning graphene and graphene-related materials and reproduction in several organisms, from *Archea* to mammals, including Human, has been provided here. What is emerging is a very interesting perspective presenting two sides of the same coin: on the one hand they could be harmful materials, with possible toxic effects on such delicate and important function; on the other hand, it is possible to speculate that the GRMs, and GO in particular, could represent a possible way to fight the problem of infertility by manipulating spermatozoa in a safer and better way.

## Figures and Tables

**Figure 1 nanomaterials-11-00547-f001:**
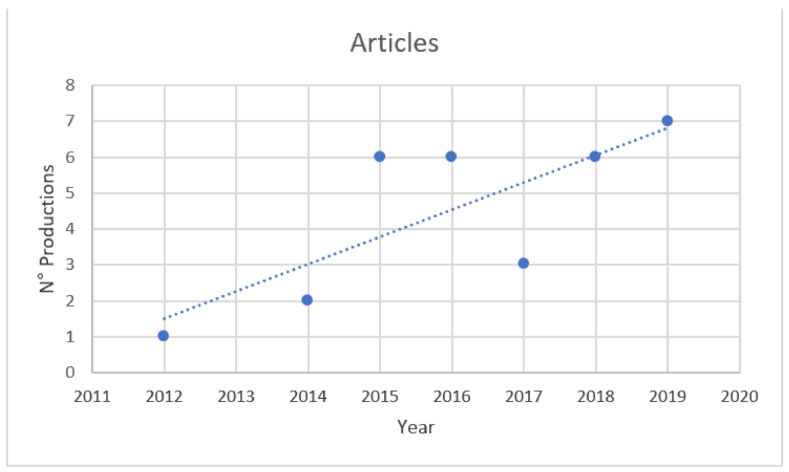
Annual scientific production. The annual growth rate shows 2012 as the starting year of publications in this field, growing exponentially since 2015 with a little collapse in 2017 and a subsequent rebound in 2018 (worthy to note that however the number of products remains very small). Blue dots correspond to the articles.

**Figure 2 nanomaterials-11-00547-f002:**
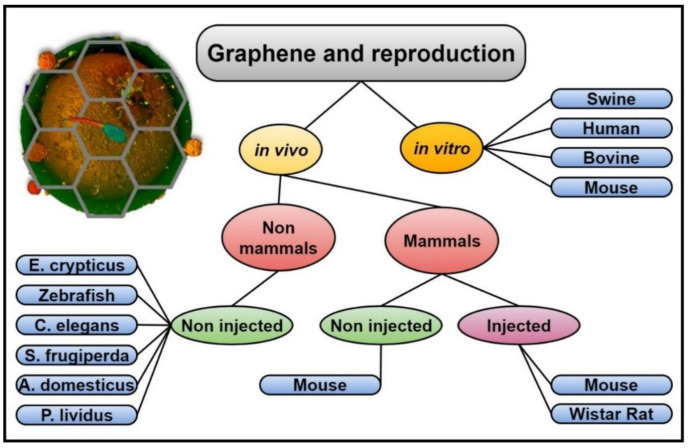
Graphene and reproduction. Graphic outline of the studies involving graphene and reproduction, either in vivo or in vitro, following different protocols and performed in different animal species.

**Table 1 nanomaterials-11-00547-t001:** Bibliometric data from the analysed documents.

Description	Results
Documents	29
Sources (Journal, Books, etc.)	23
Keywords Plus (ID)	173
Author’s Keywords (DE)	98
Period	2012–2019
Average citations per documents	19.81
Authors	176
Authors Appearances	208
Authors of single-authored documents	1
Authors of multi-authored documents	161
Single-authored documents	1
Documents per Author	0.1179
Authors per Document	5.59
Co-Authors per Documents	6.69
Collaboration Index	5.75
**Document types**	
ARTICLE	26
REVIEW	3

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
