# Peer review of "Graphene and Reproduction: A Love-Hate Relationship"

_nanomaterials, 2021, doi:10.3390/nano11020547_

Round 1

Reviewer 1 Report

Review article" Graphene and Reproduction: a love-hate relationship" deals with the topic of potential toxicity derived from intake of graphene derivatives on reproduction among different species. Authors lay a lot of weight to show the topis from its statistical perspective using scientific databases. The analysis part is followed by a basic graphene derivatives description (3 Graphene family), which can be found elsewhere. The reproduction part is basically about general toxicity, lacking the description of the effects on next generations (exception non-mammals e.g. Acheta), diseases the offspring developed, cumulative effects etc.

This article has 7 pages of statistical analysis and additional 7 pages of article data analysis (15th page is the conclusion I do not count that).

In my opinion, the second part of this article has the potential to be published. Statistical analysis should be limited to a small percentage as well as graphene family description. The title should be changed to dealing with toxicity with some focus on reproductive cels. Reproduction does not end on fertilization. Proposition e.g. "Graphene-based toxicity with a focus on reproductive cells." (more or less)

It is known that the male factor is responsible for around the 50% of the cases, and 609 among them, the 30% are due to unknown causes. Please provide a reference for this statement

Chapter 5 is one of the most important chapters considering the title. It should have more cases studied and references

Lines 590 -599 what was the effect on the offspring? The referenced article does not provide this data. At least I could not find it.  I think that graphene derivatives' effect on mammals reproduction (I do not mean only fertilization) but long term effects are still unknown, and lack of prolonged effects shows that it is too early to write reviews about it.

Some text corrections if you would like to keep the first part.

I think you can write this part without mentioning the Boolean operator in the form of a sentence and not computer code

"The following syntax was used: 96

TS = (topic 1) AND TS = (topic 2) 97

Where "TS" is the topic and "AND" is the Boolean operator 98

For the present analysis topic 1 "graphene" was used in combination with the fol-99 lowing key words as topic 2: "sperm", "spermatozoa", "human spermatozoa", "oocytes", 100 "reproduction", "fertility" and "human fertility"."

Line 102 "As a result, 31 documents…." And in Table 1 is  29 (26+3) unless it divides into further document types you did not mention

Comment to Figure 1 Number of articles is increasing, but you should also mention that the number is small.

Lines 122-125 small letters would be better, in my opinion.

Write the database source of data for Figure 3

"Figure 6. Co-authors network. The network shows that all the authors tend to form 186 highly clustered structures that do not communicate with each other, suggesting a high 187 fragmentation within the scientific community and the lack of communication among sci-188 entist (authors represented as nodes; coauthorship represented as links). The network 189 was created with Cytoscape."

To each group in Figure 6 add the name of the country above the group in the image. Does it correspond to the country's GDP spent on science? Can this plot also show nationalism in science? Does it mean that researcher cite their work inside fixed groups of collaborators and not the article of highest importance? What exactly is the source of this plot coauthorship in articles with specific keyword or citations.

As you can see my thoughts are not related to the main topic of this article. Do you want to lead me here with this analysis?  

Author Response

Reviewer 1

Review article" Graphene and Reproduction: a love-hate relationship" deals with the topic of potential toxicity derived from intake of graphene derivatives on reproduction among different species. Authors lay a lot of weight to show the topis from its statistical perspective using scientific databases. The analysis part is followed by a basic graphene derivatives description (3 Graphene family), which can be found elsewhere. The reproduction part is basically about general toxicity, lacking the description of the effects on next generations (exception non-mammals e.g. Acheta), diseases the offspring developed, cumulative effects etc.

Response: The Authors thank the reviewer 1 for his/her time and the detailed analysis of the work. Numerous corrections have been made following his/her advice and suggestions, which have helped, in our opinion, to increase the quality of this manuscript.

Unfortunately, there are no works evaluating the effects on the next generations in mammals, nor the diseases developed, etc. However, our group is making great efforts in elucidating these effects and hopefully will be available soon.

This article has 7 pages of statistical analysis and additional 7 pages of article data analysis (15th page is the conclusion I do not count that).

Response: Following the reviewers advice, the Scientometric analysis section has been reduced and simplified.

In my opinion, the second part of this article has the potential to be published. Statistical analysis should be limited to a small percentage as well as graphene family description. The title should be changed to dealing with toxicity with some focus on reproductive cels. Reproduction does not end on fertilization. Proposition e.g. "Graphene-based toxicity with a focus on reproductive cells." (more or less)

Response: Thank you for your positive comment about our manuscript. We have followed your advice and have reduced the scientometric analysis part. Moreover, we thank you for your suggestion about the title. However, we understand here reproduction as the whole field including many processes, species and cell types, but not limited to the focus on the gametes (as spermatozoa), even if we agree with you in the importance of these cells. Since this work gathers also the toxicity experiments realized on the whole individual, the term “reproductive cell” within the title could carry the reader to a misunderstanding. Instead, the presentation of graphene family has been restructured and repetitions have been erased in order to render the section more synthetic, clear and fluent.

It is known that the male factor is responsible for around the 50% of the cases, and 609 among them, the 30% are due to unknown causes. Please provide a reference for this statement

Response: Thank you for remarking this type error. We apologize for the inconveniences. The following references were added to the manuscript to support this data.

  • K. Jensen, R. Jacobsen, K. Christensen, N.C. Nielsen, E. Bostofte, Good semen quality and life expectancy: a cohort study of 43,277 men., Am. J. Epidemiol. 170 (2009) 559–65. Doi:10.1093/aje/kwp168.
  • D. Ring, A.A. Lwin, T.S. Köhler, Current medical management of endocrine-related male infertility., Asian J. Androl. 18 (2016) 357–63. Doi:10.4103/1008-682X.179252.
  • R. Winters, T.J. Walsh, The epidemiology of male infertility., Urol. Clin. North Am. 41 (2014) 195–204. Doi:10.1016/j.ucl.2013.08.006.

Chapter 5 is one of the most important chapters considering the title. It should have more cases studied and references

Response: Thank you for expressing this opinion. We totally agree with you: this is on of the most important chapters and it should have more cases and references, but, however, and to our knowledge, there is no more articles regarding this aim, as there are no articles regarding the effects of graphene (and derivatives) on the progeny. Actually, this review manuscripts born with the intention of raising awareness about two main issues:

-The need of evaluating the effects of graphene and derivatives not only in the individual but also on the progeny (when exposed to the materials, either the individual or the gametes) and

-The importance of considering the potential positive effects that could be derived from the materials interaction with the biological system, and, in this case, with spermatozoa. Although several researchers have searched for the negative outcomes, only few of them have focused their studies in taking advantage from the potential positive effects, which might be derived from physical or chemical interactions.

Lines 590 -599 what was the effect on the offspring? The referenced article does not provide this data. At least I could not find it.  I think that graphene derivatives' effect on mammals reproduction (I do not mean only fertilization) but long term effects are still unknown, and lack of prolonged effects shows that it is too early to write reviews about it.

Response: We totally agree with the Reviewer about this issue. We do not know the effects on the offspring by now, but experiments are being performed to elucidate the potential long-term effects. To date, and thanks to the experiments realized using the mouse model (Bernabò et al. Front. Bioeng. Biotechnol 2020) we can assure the obtention of a higher number of fertilized oocytes and thus a higher number of embryos which after implantation in foster mothers gave birth to a phenotypically healthy offspring.

Reference:

Bernabò N, Valbonetti L, Raspa M, Fontana A, Palestini P, Botto L, Paoletti R, Fray M, Allen S, Machado-Simoes J, Ramal-Sanchez M, Pilato S, Scavizzi F and Barboni B (2020) Graphene Oxide Improves in vitro Fertilization in Mice With No Impact on Embryo Development and Preserves the Membrane Microdomains Architecture. Front. Bioeng. Biotechnol. 8:629. doi: 10.3389/fbioe.2020.00629

Some text corrections if you would like to keep the first part.

I think you can write this part without mentioning the Boolean operator in the form of a sentence and not computer code

"The following syntax was used: 96

TS = (topic 1) AND TS = (topic 2) 97

Where "TS" is the topic and "AND" is the Boolean operator 98

For the present analysis topic 1 "graphene" was used in combination with the fol-99 lowing key words as topic 2: "sperm", "spermatozoa", "human spermatozoa", "oocytes", 100 "reproduction", "fertility" and "human fertility"."

Response: Following your suggestion, this part was suppressed, and the whole section was reduced.

Line 102 "As a result, 31 documents…." And in Table 1 is  29 (26+3) unless it divides into further document types you did not mention.

Response: Thank you again for remarking this typo. Actually, the number of documents is 31 (29 research articles + 3 reviews) but the system sensibility reached only 29 (26+3). Two manuscript could not be included in the analysis (Kong et al. 2019 and Bernabò et al. 2020) because were published after the downloading of the data. This part was modified within the text for a better understanding of the reader.

Comment to Figure 1 Number of articles is increasing, but you should also mention that the number is small.

Response: it was mentioned following your advice (Fig.1, L15 and L120)

Lines 122-125 small letters would be better, in my opinion.

Response: it was changed, following your suggestion.

Write the database source of data for Figure 3

Response: as for the other figures, the database used was the Advanced Search Function of Web of Science Database (WoS). However, Figure 3 was eliminated from the manuscript in an effort to reduce the Scientometric analysis section.  

"Figure 6. Co-authors network. The network shows that all the authors tend to form 186 highly clustered structures that do not communicate with each other, suggesting a high 187 fragmentation within the scientific community and the lack of communication among sci-188 entist (authors represented as nodes; coauthorship represented as links). The network 189 was created with Cytoscape."

To each group in Figure 6 add the name of the country above the group in the image. Does it correspond to the country's GDP spent on science? Can this plot also show nationalism in science? Does it mean that researcher cite their work inside fixed groups of collaborators and not the article of highest importance? What exactly is the source of this plot coauthorship in articles with specific keyword or citations.

As you can see my thoughts are not related to the main topic of this article. Do you want to lead me here with this analysis? 

Response: We thank Rev 1 for this interesting question. The purpose of Figure 6 (now Figure 2) and its analysis carried out here is to study the interconnection among the different groups involved in Graphene and reproduction research. This, in our opinion, leads to interesting results, showing as it does not exist a big scientific community focusing their attention on this topic, but sparse small communities study it.

We already found this pattern in other different contexts (Bernabò et al., PLOS ONE 2017) and here it could be due to the very recent onset of the research activity focused on these materials and their effect on health.

The identification and labelling of the research groups was out of the scope of this analysis since it would need a more specialist and focused approach. Indeed, several factor must be considered:

  • Each group could be constituted by researchers from different Countries
  • Researchers from the same Country could belong to different institutions
  • Possibly the same Author could belong to two or more institutions, and could move from different institutions within the time
  • It is impossible to identify a defined cause-effect relationship between the research expense (e.g., expressed as GDP) at national or international level and the research on this topic (it could be funded by industrial research programmes, by national and international Agencies, etc…)(Bernabò et al., Scientometrics 2016). A study on European (we are in Italy) or world-wide dynamic of funding on graphene and graphene-related material could be very intriguing and interesting, but however it is out of the scope of this manuscript.

References:

  • Bernabò N, Ciccarelli R, Greco L, Ordinelli A, Mattioli M, et al. (2017) Scientometric study of the effects of exposure to non-ionizing electromagnetic fields on fertility: A contribution to understanding the reasons of partial failure. PLOS ONE 12(12): e0187890. https://doi.org/10.1371/journal.pone.0187890
  • Bernabò, N., Greco, L., Mattioli, M. et al. A scientometric analysis of reproductive medicine. Scientometrics 109, 103–120 (2016). https://doi.org/10.1007/s11192-016-1969-3

Reviewer 2 Report

Having a topical subject, the present paper  elaborated after the rules of scientometric review deserves publication after the  folowing modification in the frame of a major revision  :

  1. Is a need for better material organization being a large part devoted not to graphene merits and demerits and only to the description of the way to elaborate a scientometric review. Of course a review has to contain such a part, but the present manuscript has more than a third with such a part. However the ratio between material in subchapters has to be introduced in  a new distribution
  2. There are seven 7 figures and only one is devoted to the paper topic, the other being about review elaboration
  3. For all figures is not clearly presented the level of their original character

Author Response

Reviewer 2

Having a topical subject, the present paper  elaborated after the rules of scientometric review deserves publication after the  folowing modification in the frame of a major revision  :

1. Is a need for better material organization being a large part devoted not to graphene merits and demerits and only to the description of the way to elaborate a scientometric review. Of course a review has to contain such a part, but the present manuscript has more than a third with such a part. However the ratio between material in subchapters has to be introduced in  a new distribution

Response: First of all, we would like to thank Reviewer 2 for his/her time for revision and the suggestion made to this work. Following your advices, we have modified the manuscript, which we consider that has considerably improved now. For instance, Scientometry analysis section was considerably reduced, following Reviewers advice, to remark the importance of the other sections. Finally, carbon-derived materials have been presented in a new distribution that considers real subchapters rather than using the simple, and disharmonious with the editing of the manuscript, alphabetical list. Moreover, repetitions have been erased in order to render this part of the manuscript easy to read and fluent.

2. There are seven 7 figures and only one is devoted to the paper topic, the other being about review elaboration

Response: Thank you for your comment. Following your advice, we have reduced the number of figures regarding the scientometric analysis, in an effort to give more protagonist to the paper topic and avoid misunderstanding to the reader.

3. For all figures is not clearly presented the level of their original character

Response: Thank you for your comment. We have eliminated several figures corresponding to the scientometric analysis. We apologize for the inconveniences and hope that the analysis could be considered clearly presented now.

Round 2

Reviewer 1 Report

The answers provided by the authors suit me fine, and I agree with them. I see that authors tend to scientometric analysis, which could be a topic of the next article. In general thorough and broad analysis of scientist interdependencies with a clear focus on problems like self-citations and citing articles of so-called “friends” instead of the best articles could be an essential contribution to science development. Although it should be a totally separate topic dealing with sociology, politic of science, forced parametric improvements (e.g., improving H index).

The shortcuts authors did are fine, but still, there is a problem with the readability of fig 2. Names are not visible for a reader that are only groups of not labelled, interconnected balls.  You have two options correct that or remove this image leaving in the text the description only. Both solutions are fine with me.

Description of carbon family is shorter, so it is fine

Decide what to do with Fig 2, and I have no more objections. Propose to publish after this correction will be done.

Author Response

Thank you for your comments and your positive review. Actually, we agree with Rev 1 about the readibility of the figure 2 (clusters with the authors names), thus we decided to remove this figure following Rev 1 advice. 

Reviewer 2 Report

 Having a better organization of subchapter dimension and of figures number as well the new manuscript version could be published in present form

Author Response

Thank you for your positive review.